# A Harvester with a Helix S-Type Vertical Axis to Capture Random Breeze Energy Efficiently

**DOI:** 10.3390/mi14071466

**Published:** 2023-07-21

**Authors:** Chao Zhang, Boren Zhang, Jintao Liang, Zhengfeng Ming, Tao Wen, Xinlong Yang

**Affiliations:** School of Mechano-Electronic Engineering, Xidian University, Xi’an 710071, China; 21041212037@stu.xidian.edu.cn (B.Z.); jtliang@xidian.edu.cn (J.L.); zfming@xidian.edu.cn (Z.M.); wentao@xidian.edu.cn (T.W.); 22041212928@stu.xidian.edu.cn (X.Y.)

**Keywords:** helix s-type vertical axis, natural breeze energy, lower startup speed, improved energy retention

## Abstract

Breeze energy is a widely distributed renewable energy source in the natural world, but its efficient exploitation is very difficult. The conventional harvester with fixed arm length (HFA) has a relatively high start-up wind speed owing to its high and constant rotational inertia. Therefore, this paper proposes a harvester with a helix s-type vertical axis (HSVA) for achieving random energy capture in the natural breeze environment. The HSVA is constructed with two semi-circular buckets driven by the difference of the drag exerted, and the wind energy is transferred into mechanical energy. Firstly, as the wind speed changes, the HSVA harvester can match the random breeze to obtain highly efficient power. Compared with the HFA harvester, the power coefficient is significantly improved from 0.15 to 0.2 without additional equipment. Furthermore, it has more time for energy attenuation as the wind speeds dropped from strong to moderate. Moreover, the starting torque is also better than that of HFA harvester. Experiments showed that the HSVA harvester can improve power performance on the grounds of the wind speed ranging in 0.8–10.1 m/s, and that the star-up wind speed is 0.8 m/s and output peak power can reach 17.1 mW. In comparison with the HFA harvester, the HSVA harvester can obtain higher efficient power, requires lower startup speed and keeps energy longer under the same time. Additionally, as a distributed energy source, the HSVA harvester can provide a self-generating power supply to electronic sensors for monitoring the surrounding environment.

## 1. Introduction

With the continuous development of intelligence and information technology, the world has entered the intelligent era [1], which requires clean, renewable and distributed energy to satisfy the huge energy demand of trillions of sensors worldwide [2,3,4,5]. Therefore, it is difficult for conventional methods of energy supply to meet these needs, which puts forward new requirements for energy supply. For a long time, the power supply of distributed sensors depended on power grids or batteries, which has brought some serious negative effects on the natural environment [6,7,8,9,10]. More advanced technologies need to be developed for distributed energy. Wind energy has great potential to solve the above problems due to its pollution-free and extensive availability virtues [11,12,13,14,15,16,17,18].

However, collection of low-speed wind energy still faces some challenges, such as low utilization rate and huge energy loss [19,20,21]. It is notable that breeze energy has huge reserves all over the world, and rotational energy harvesters can effectively convert mechanical energy into electrical energy by coupling and electrostatic induction. Therefore, the rotational energy harvester is well-suited for distributed energy collection. Currently, the performance of rotational energy harvesters can be improved through their mechanisms, circuits, materials and theoretical basis [22,23,24,25,26,27]. However, breeze energy has the characteristics of randomness, volatility and instability, and driving torque is one of the main parameters affecting the performance of harvesters [28,29,30,31]. Ref. [32] proposes a self-power-supplying device which is based on triboelectric nanogenerators and an electromagnetic generator. The energy harvesting and sensing device based on an electromagnetic triboelectric hybrid generator can indirectly measure wind speed. However, this device has a large and constant rotational inertia which gives it a relatively high start-up wind speed. Ref. [14] designs a triboelectric electromagnetic hybrid nanogenerator for self-powered sensors. Nevertheless, the driving arm length of the harvester is fixed, which makes it difficult to adapt to the changing wind energy in the natural environment. Therefore, if the arm length of the harvester is automatically adjusted by the mechanical structure, the output power can be increased efficiently [4,33,34]. Furthermore, the decay time increases continuously along with the recovered energy as the wind speed decreases gradually [35,36].

Therefore, if the mechanical structure of the wind cup is changed to match the input energy, the output power could be increased substantially [37]. The Savonius wind harvester is constructed by two semi-circular buckets, and it has a better starting torque at low wind speeds [38,39]. Ref. [40] investigated the effects on the performance of a deflecting plate, which they have applied for improving the performance of the Savonius wind harvester. Ref. [41] proposed a wind-driven harvester to prevent the negative torque opposite to the rotation of the harvester. Ref. [42] improved the performance of a harvester using a guide-box tunnel. Furthermore, ref. [43] continuously increased decay time with respect to recovered energy as the wind speed decreases gradually. Therefore, the structural characteristics of the harvester should be designed to automatically adapt to the fluctuations in the breeze environment. 

In this paper, a harvester with a helix s-type vertical axis (HSVA) is developed for efficiently collecting random breeze energy. The mechanical structure of the HSVA harvester can achieve dynamic matching between generation unit and external breeze energy. When the external wind speed changes, the HSVA harvester can enhance the power generation efficiency of the generator to decrease the power error by changing its own mechanical structure to match wind speed. In the random breeze environment, experimental results show that the maximum energy conversion efficiency of the HSVA harvester is higher than that of harvesters with fixed arm length (HFA). When the wind speed decreased significantly, in comparison with the energy decay time, the HSVA harvester can obtain lower startup speed and longer energy storage time than the HFA harvester. Additionally, the structure of the HSVA harvester has a strong protective effect as the wind speed rises to 10 m/s. 

## 2. Results and Discussion

### 2.1. Structure Design

Figure 1a shows the basic structure of the three-wind-cup harvester. The structure of the HSVA harvester is displayed in Figure 1b, including wind cup with helix s-type vertical axis harvester, generation unit, and shell. The HSVA harvester is constructed by two semi-circular buckets. The HSVA harvester is pushed by the wind on semicircular vanes, resulting from the difference of the drag exerted. Therefore, the breeze energy is transferred into mechanical energy by the HSVA harvester rotates. In addition, the HSVA harvester has a better starting torque at lower wind speeds in the random breeze environment. Figure 1c shows a photograph of the as-fabricated HSVA harvester, and Figure 1d shows the top view of the HSVA harvester. 

Figure 1b shows a single screw harvester with two blades, which can be defined as a curve generated by marking vertical motion at a constant angle on a rotating cylinder. The inner and outer edges are twisted at 90 degrees through a quarter-pitch turn. The blade maintains its semicircular cross-section from the bottom (0°) to the top (90°). In the paper, the two blades are called the helix s-type vertical axis.

### 2.2. Energy in the Wind

For an air stream flowing through an area *A*, the mass flow rate is ρAv, and therefore the power is
(1)$$P=\frac{1}{2}\rho Av^{3}$$

*P* is the active power and is also known as the energy flux or power density of air. ρ is the air density, *A* is the projected area and *v* is the air speed at the front of the harvester. The power coefficient (*C_p_*) indicates
(2)$$C_{p}=\frac{P_{s}}{P}$$

The ratio of shaft power (*P_s_*) is calculated from brake torque and rotating speed
(3)$$P_{s}=\frac{1}{2}\rho A\left[C_{T1}{\left(v-u\right)}^{2}-C_{T2}{\left(v+u\right)}^{2}\right]u=T\omega$$

$C_{T1}$ is the resistance coefficient of downwind blades, *u* is the tip velocity, $C_{T2}$ is the resistance coefficient of upwind blades and ω is the rotating speed. Two situations are considered. If *u* = 0, meaning that the harvester has zero rotating speed, there is no power output. If *u* = *v*, this means the top of the blade moves with the wind speed and the power output is negative. Therefore, there must be an optimal tip speed ratio (TSR) between 0 and 1 in order to achieve the maximum coefficient of the harvester.

TSR is given by
(4)$$\mathrm{TSR}=\frac{u}{v}$$
and *C_T_* is
(5)$$C_{T}=\frac{4T}{\rho u^{2}d^{2}H}$$

### 2.3. Mechanism of the Generation Unit

The generation unit consists of the stator and the rotor. Figure 2 shows the electromagnetic induction effect that the generation unit works on. When a magnet fixed on the rotor rotates towards a fixed coil, the magnetic flux via the coil increases gradually. The induced current is generated in the coil. However, the subsequent magnetic field hinders the decline of magnetic flux (I). As the rotor rotates, the magnet is farther away from the fixed coil. This phenomenon makes the magnetic flux through the coil decline. Therefore, in the coil, the current flows in the opposite direction, supplementing the reduced magnetic flux (II). Additionally, by continuous rotation, the adjacent magnet deviates from the coil, and the current flows again in alternating directions (III and IV). By this means, under the continuous rotation of the rotor, the magnetic field changes direction alternately, and then the periodic alternating current is generated. 

### 2.4. Research Method

The standard k- ε turbulence model is used in conjunction with the logarithmic surface function for turbulence analysis. Some variables have been solved by using this program, including the momentum equation, *x*, *y* and *z* components of velocity, turbulent kinetic energy (*k*) and dissipation rate of turbulent kinetic energy (*E*). All these equations are formulated by using iterative methods to provide each equation at the center point of the element, and a quadratic interpolation method with high reliability level is adopted. In addition, in order to maintain the availability and quality protection of the pressure area during each iteration, pressure correction is addressed. Moreover, the SIMPLE analysis algorithm is used to calculate pressure and velocity distribution.

The mass and momentum protection equations have been used in this program, which can be written for compressible and incompressible steady flows in Cartesian tensor rotation as follows:

The continuity equation,
(6)$$\frac{\partial }{\partial x_{j}}\left(\rho \cdot u_{j}\right)=0$$
the momentum common equation,
(7)$$\frac{\partial }{\partial x_{j}}\left(\rho \cdot u_{j}\cdot u_{i}-\tau_{ij}\right)=\frac{\partial p}{\partial x_{j}}+S_{i}$$

*x_i_* is the Cartesian coordinate (*j* = 1, 2, 3), *u_i_* is the absolute velocity components in the direction of *x_i_*, *P* is the Piezometric pressure and $\tau_{ij}$ is the stress tensor component.

The Turbulent kinetic energy (*k*)
(8)$$\frac{\partial }{\partial t}\left(\rho k\right)+\frac{\partial }{\partial x_{j}}\left(\rho u_{j}k-\frac{u_{eff}}{\sigma_{k}}\cdot \frac{\partial k}{\partial x_{j}}\right)=u_{t}s_{ij}\frac{\partial u_{i}}{\partial x_{j}}-\frac{2}{3}\left(u_{t}\frac{\partial u_{i}}{\partial x_{i}}+\rho k\right)\frac{\partial u_{i}}{\partial x_{i}}-\rho \varepsilon$$
the Dissipation rate of turbulent kinetic energy (ε)
(9)$$\frac{\partial }{\partial t}\left(\rho \varepsilon \right)+\frac{\partial }{\partial x_{j}}\left(\rho u_{j}\varepsilon -\frac{u_{eff}}{\sigma_{\varepsilon }}\cdot \frac{\partial \varepsilon }{\partial x_{j}}\right)=C_{1}f_{1}\frac{\varepsilon }{k}\left[u_{t}s_{ij}-\frac{2}{3}\left(u_{t}\frac{\partial u_{i}}{\partial x_{i}}+\rho k\right)\frac{\partial u_{i}}{\partial x_{i}}\delta_{ij}\right]-C_{2}f_{2}\rho \frac{\varepsilon^{2}}{k}-C_{3}\rho \varepsilon \frac{\partial u_{i}}{\partial x_{i}}$$

In these equations, the $C_{\mu }$, $\sigma_{k}$, $\sigma_{\varepsilon }$, $C_{1\varepsilon }$ and $C_{2\varepsilon }$ are the empirical constants for the turbulence model, which are equal to 0.09, 1.0, 1.3, 1.44 and 1.92, respectively.

#### 2.4.1. Effect of Twist Angle θ

Figure 3 shows the performance curve of the spiral-S wind harvester of different twisted angles. It can be seen that the energy conversion efficiency is significantly higher than that of 90°, 270° and 360°, with the maximum value of 0.175 at TSR = 0.72.

#### 2.4.2. Effect of Inner Plates Number N

Figure 1b shows that the inner plates increase the pressure of the harvester on convex wind backside and concave surface. This shows that the inner plates increase the height of the concave blade units, thereby increasing the torque. In addition, this increases the replenishing breeze energy of the concave surface airflow on the air side of the convex surface, and the positive torque also increases.

Moreover, it also increases the replenishing breeze energy of concave plane air on the side of the convex surface, and the negative torque has decreased. In addition, the inner plates can improve the conversion of the harvester, and also can improve the strength of the blade structure and the ability to resist strong wind. Ref. [39] concluded that a harvester with 4–5 inner plates can obtain the highest energy conversion efficiency; at this time, the conversion rate is 0.187 and TSR = 0.73. 

## 3. Performance Comparison

In order to prove that the HSVA harvester achieves a lower start-up wind speed, Figure 4a,b compared the HFA harvester and the HSVA harvester, and the variation trend of rotation speed and output power are shown respectively in different speeds. The peak rotating speed of the HSVA harvester is 634 rpm at 10.1 m/s, and the maximum peak power of the HSVA harvester is 17.1 mW. These performance values are far higher than those of the HFA harvester. In addition, because the HSVA harvester is easily driven, this study opted to use the helix s-type vertical axis harvester to allow the HSVA harvester to adjust to the typical range of wind speeds found in the natural environment. In a moderate breeze environment, the start-up wind speed of the HSVA harvester is 0.8 m/s, which is lower than the start-up wind speed of the HFA harvester (1.2 m/s).

In Figure 4b, as the wind speed rises from 0 m/s to 3.6 m/s, the output power of the two generators is basically not much different, because the structural characteristics of the HSVA harvester have not been fully utilized at a moderate breeze. However, as wind speed continues to rise, in comparison with the HFA harvester, the structural characteristics of the HSVA harvester adjust to strong speeds, but those of the HFA harvester do not. As the wind speed rises from 3.6 m/s to 10.1 m/s, the power growth rate of the HSVA harvester is 400%, which is 2.95 times that of the HFA harvester (5.8 mW). Therefore, the HSVA harvester is more efficient than the HFA harvester. 

Table 1 shows the relationship between wind speed and output power of two different harvesters. It is proven that the HSVA harvester has better power conversion efficiency under different random wind speeds after the two harvesters start. As the wind speed increases gradually, the gap in conversion efficiency widens between the two harvesters.

Moreover, to further prove the application capability, the paper simulates an environment with increasing and decreasing wind speed, and the output power of the HFA and HSVA harvester is compared. Figure 5 shows the overall experimental environment. In Figure 5a,b, the output voltage is compared between the two generators as the wind speed is raised from 0 m/s to 4.1 m/s. Obviously, as the wind speed gradually increases to the starting wind speed, the HSVA harvester is the first to start, achieving energy conversion. Furthermore, as the wind speed gradually increases, the HSVA harvester achieves a higher voltage level. The output peak voltage can reach 2.6 V at the wind speed of 10.1 m/s, which is 1.73 times higher as compared with the corresponding output of the HFA harvester. Moreover, the comparative rising trend of voltage change is shown in Figure 5c,d: as the wind speed starts from 1.2 m/s, the interval is 20 s, the wind speed rises to 2.4 m/s, and 20 s later, the wind speed rises to 4.1. It is noted that much higher voltage can be achieved for the HSVA harvester at each stage of wind speed change. Therefore, the HSVA harvester obtains better energy conversion efficiency. On the contrary, the comparative downtrend of power change is shown in Figure 3e,f, when wind speed drops from 4.1 to 1.2 at the same interval. As wind speed increases or decreases, the wind cup captures the random breeze energy of the environment directly, and the HSVA harvester can automatically adjust to typical speeds of different winds. The energy conversion efficiency of the HSVA harvester is much higher than that of the HFA harvester for achieving random breeze energy capture.

Figure 6 shows the power decay time when the wind speed rises from 0 m/s to 4.1 m/s for two harvesters. The HFA harvester is considered to better prove the effectiveness of the HSVA harvester. In Figure 6a,b, the application experiments proved that the decay time of the HSVA harvester was 44.6 s higher than that of the HFA harvester (21.4 s).

Therefore, this proves the characteristic of high energy retention for the HSVA harvester. Moreover, in environments of strong breezes, the function of the helix s-type vertical axis can protect the structure of the HSVA harvester, avoiding damage. 

## 4. Applications

Because wind energy has the characteristics of randomness, volatility and intermittency, a bridge rectifier and a storage capacitor are applied in the energy management unit, which converts the AC outputs from the HSVA harvester to DC outputs. Furthermore, the generated power can be accumulated in the capacitor for actual utilization. 

In Figure 7a, under the wind speed of 4.1 m/s, the 100 µF capacitor is charged to 2.8 V and 2.24 V, respectively, by the HSVA and the HFA harvesters. In Figure 7b, when the wind speed is 4.1 m/s, the voltages are charged with different capacitors. The 100 µF capacitor is charged to 2.8 V within about 4.9 s. The voltage of a larger 470 µF capacitor can gradually reach the saturated value of 2.7 V within about 15 s. As a result, the superior charging capability is demonstrated by the HSVA harvester. In Figure 7c, as the wind speed rises from 1.2 m/s to 4.5 m/s, the rotor speed grows linearly from 85 rpm to 300 rpm. Similarly, the voltage signal is obtained, and its frequency also grows linearly from 17 Hz to 60 Hz. Moreover, the linear relationship between them makes the HSVA harvester become a self-sufficient power source.

Additionally, in Figure 7d,e, the HSVA harvester is applied to power a wireless hygrothermograph, which can sense the surrounding temperature as well as humidity, and the data is transmitted to a smart phone through Bluetooth technology. Furthermore, the rectified electrical energy is stored in a 2200 µF capacitor that provides a relatively stable power source for the sensors. Moreover, about 11 s is taken for the HSVA harvester to charge the capacitor to 2.8 V at wind speeds of around 4.2 m/s. 

It can be found that there is a fast drop in the voltage of the capacitor when the hygrothermograph is turned on. The reason why this phenomenon occurs is that the system initialization and the wireless connection establishment cause large energy dissipation. In addition, the storage capacitor can approach a stable voltage of about 1.8 V gradually after the initialization. As a result, sufficient electrical energy can be offered by the HSVA harvester to power the wireless hygrothermograph and maintain its continuous operation.

## 5. Conclusions

This study proposes a HSVA harvester solution to the mismatch between the HFA harvester and input energy in the breeze environment. Firstly, the HSVA harvester can match the random wind to achieve highly efficient power in the natural breeze environment. Secondly, the HSVA is constructed by two semi-circular buckets driven by the difference of the drag exerted, and the proposed harvester achieves a low start-up wind speed of 0.8 m/s. Finally, after the wind stops, the HSVA harvester can work for a longer duration than the HFA harvester. The experimental results verify the effectiveness of the proposed HSVA harvester. The output peak power can reach 17.1 mW at a wind speed of 10.1 m/s, which is 2.95 times higher as compared with the corresponding output of the HFA harvester. Furthermore, as the wind speed gradually increases, the proposed HSVA harvester achieves a low start-up wind speed of 0.8 m/s. Moreover, experiments showed that the operating time of the HSVA harvester was 44.6 s longer than that of the HFA harvester after the wind stopped from 4.5 m/s. The HSVA harvester is well suited for the exploitation of natural random wind that normally has weak driving force, varied speed, and intermittent availability. Therefore, the HSVA harvester can serve as a green energy source for sensors to monitor the environment. 

## Figures and Tables

**Figure 1 micromachines-14-01466-f001:**
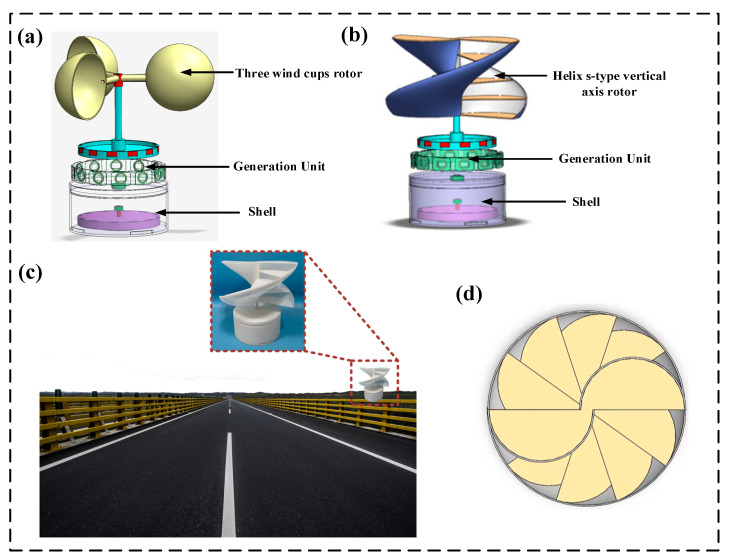
Schematic showing the structure of harvesters: (**a**) basic structure of the HFA harvester (**b**) basic structure of the HSVA harvester (**c**) the HSVA harvester application by the road (**d**) top view of the HSVA harvester.

**Figure 2 micromachines-14-01466-f002:**
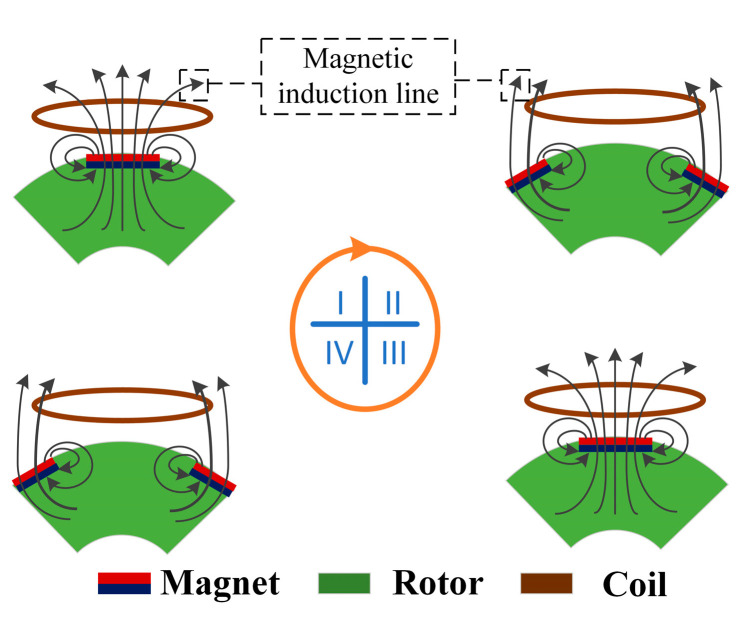
Electromagnetic induction effect at four typical situations.

**Figure 3 micromachines-14-01466-f003:**
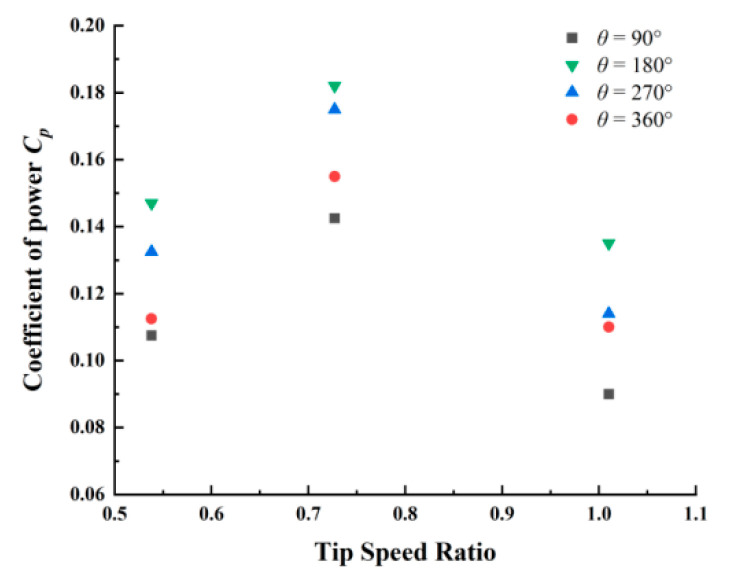
Effect of helical angle on power coefficient.

**Figure 4 micromachines-14-01466-f004:**
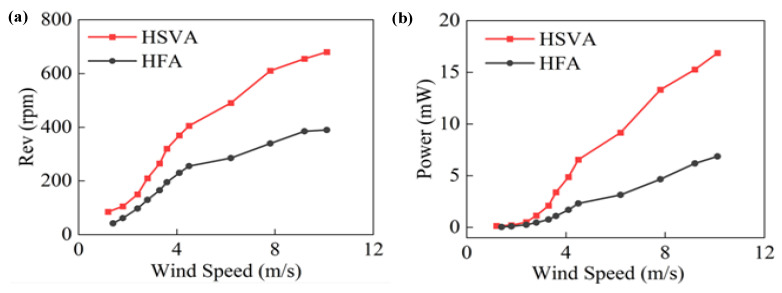
Output performance of the two harvesters: (**a**) Rotating speed comparison in different wind speed (**b**) output power comparison in different wind speed.

**Figure 5 micromachines-14-01466-f005:**
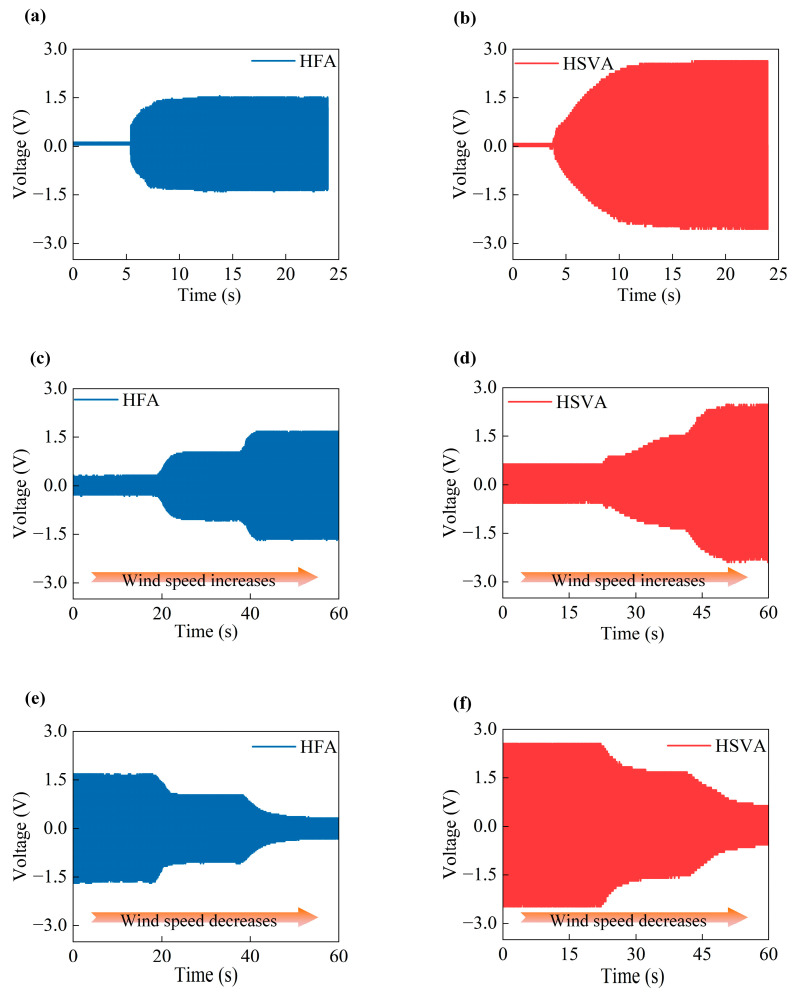
Output voltage performance between (**a**) the HFA harvester out voltage from 0 m/s to 4.1 m/s (**b**) the HSVA harvester out voltage when wind speed is risen from 0 m/s to 4.1 m/s (**c**) the HFA harvester out voltage when wind speed is risen from 1.2 m/s to 2.4 m/s to 4.1 m/s (**d**) the HSVA harvester out voltage when wind speed is risen from 1.2 m/s to 2.4 m/s to 4.1 m/s (**e**) the HFA harvester out voltage when wind speed is risen from 4.1 m/s to 2.4 m/s to 1.2 m/s (**f**) the HSVA harvester out voltage when wind speed is risen from 4.1 m/s to 2.4 m/s to 1.2 m/s.

**Figure 6 micromachines-14-01466-f006:**
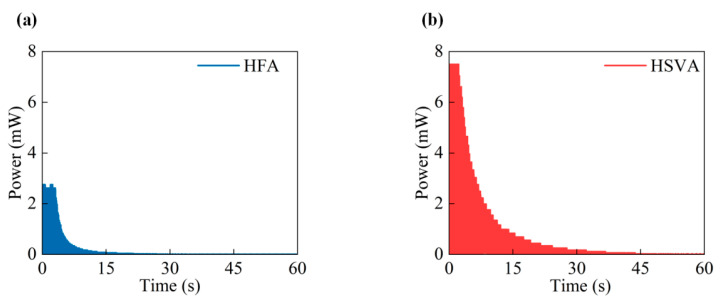
Power decay time when wind speed rises from 4.1 m/s to 0: (**a**) Power decay time of the HFA harvester (**b**) Power decay time of the HSVA harvester.

**Figure 7 micromachines-14-01466-f007:**
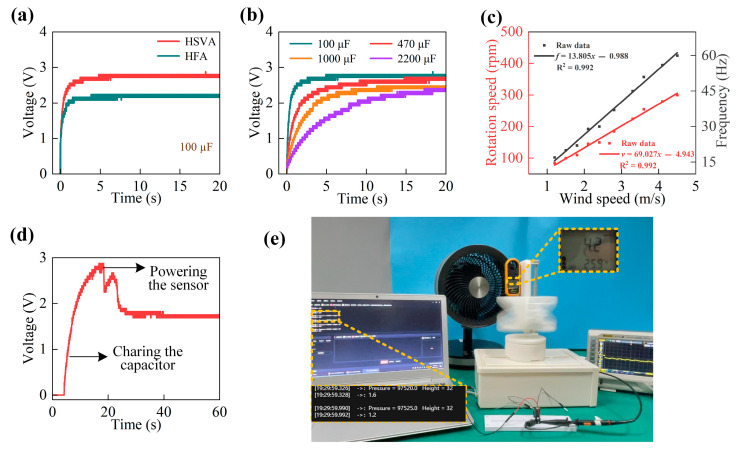
Applications of the HSVA harvester: (**a**) Voltage value curves for a 100 µF capacitor (**b**) Voltage value curves for different capacitors (**c**) Rotor speed and output frequency of the HSVA harvester (**d**) Picture of a self-sufficient sensor powered by the HSVA harvester (**e**) Picture of a wireless hygrothermograph powered by the HSVA harvester.

**Table 1 micromachines-14-01466-t001:** Relationship between wind speed and output power of two different harvesters.

Wind Speed	Out Power (HFA)	Out Power (HSVA)
2.9 m/s	0.9 mW	1.8 mW
3.7 m/s	1.1 mW	2.6 mW
4.5 m/s	1.6 mW	4.9 mW
5.5 m/s	2.9 mW	7.5 mW
7.0 m/s	3.3 mW	11.8 mW
8.1 m/s	4.0 mW	13.7 mW
10.1 m/s	6.7 mW	17.1 mW

## Data Availability

Data underlying the results presented in this paper are not publicly available at this time but may be obtained from the authors upon reasonable request.

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
