# Peer review of "A Harvester with a Helix S-Type Vertical Axis to Capture Random Breeze Energy Efficiently"

_micromachines, 2023, doi:10.3390/mi14071466_

Round 1
Reviewer 1 Report
This paper proposed a harvester with a helix s-type vertical axis to catch the wind energy. Through the structure design, this harvester gains 10.1 mW at the wind speed of 10.1 m/s, which is larger than the conventional configuration energy harvester. However, the explanation of the energy conversion process is lack. In addition, the paper writing does not meet the requirement of the journal. I think that this article cannot be accepted and major revision is needed.
Q1: The process of energy conversion should be a core part of an article related to the wind energy harvester. The detailed mechanism of the generation unit should be explained on the Page 3.
Q2: Please check the figures in the article carefully. The Figure 7 is very unclear. Please replace it with the high-resolution image.
Q3: The effect of the inner plates number is discussed in this article in Page 5, while the energy harvest performance of the helix s-type vertical axis configuration energy harvester without inner plates should be added.
Some sentences do not meet the requirement of academic writing in a qualified academic paper. The grammar mistakes and confusion are marked red.
On Page 1 Line 43:
[32] proposed a self-power supply device which is base on triboelectric nanogenerators and electromagnetic generator.
On Page 2 Line 47:
[14] designed the triboelectric electromagnetic hybrid nanogenerator for self-powered sensors.
On Page 2 Line 68:
When the external wind speed changes, the HSVA harvester can enhance the power generation efficiency of the generator to reduce the gap between actual and desired power by changing its own mechanical structure to match wind speed,
On Page 2 Line 75:
Additionally, the structure of the HSVA harvester has a strong protective 75 effect as the wind speeds rose 10.1m/s.
On Page 5 Line 170:
In addition, it also increases the replenishing breeze energy of the concave surface airflow on the side air side of the convex surface, and the positive torque has also increased.
On Page 6 Line 190:
In a moderate breeze environment, the star-up wind speed of the HSVA harvester is 0.8m/s, are smaller than those when the HFA harvester is driven (1.2m/s).
On Page 6 Line 211:
Therefore, as wind speed increased or decreased, the wind cup captures the random breeze energy of the environment directly, and the HSVA harvester can automatically adjust to typical of different winds.
On Page 8 Line 231:
Moreover, as in environment of strong breeze, the function of helix s-type vertical axis can protect the structure of the HSVA harvester avoiding damage.
On Page 8 Line 239:
On Fig. 7b, when wind speed is 4.1 m/s, the voltage trends are charged with different capacitors.
On Page 9 Line 253:
Additionally, in Fig. 7d and 7e, the HSVA harvester is applied to power a wireless hygrothermograph, which can sense the surrounding temperature as well as humidity, and the data can be send to the smart phone though the bluetooth technology.
On Page 9 Line 275:
the HSVA harvester is well suited for the exploitation of natural random wind that normally has weak driving force, varied speed, and intermittent availability.
Author Response
Reviewers’ Comments:
Reviewer: 1
This paper proposed a harvester with a helix s-type vertical axis to catch the wind energy. Through the structure design, this harvester gains 10.1 mW at the wind speed of 10.1 m/s, which is larger than the conventional configuration energy harvester. However, the explanation of the energy conversion process is lack. In addition, the paper writing does not meet the requirement of the journal. I think that this article cannot be accepted and major revision is needed.
- The process of energy conversion should be a core part of an article related to the wind energy harvester. The detailed mechanism of the generation unit should be explained on the Page 3.
Response:
In order to better explain the structure and mechanism of generation units, we have added Section 2.3 and Figure 2. The mechanism has been explained in Section 2.3.
- Please check the figures in the article carefully. The Figure 7 is very unclear. Please replace it with the high-resolution image.
Response:
We have carefully checked throughout all figures. The unclear images have been replaced in Figure.7.
- The effect of the inner plates number is discussed in this article in Page 5, while the energy harvest performance of the helix s-type vertical axis configuration energy harvester without inner plates should be added.
Response:
The inner plates of the HSVA harvester has been added in the revised manuscript.
- Some sentences do not meet the requirement of academic writing in a qualified academic paper. The grammar mistakes and confusion are marked red.
Response:
We have carefully checked throughout the manuscript to correct grammatical errors. All those mistakes have been listed as following:
- On Page 1 Line 43:“[32] proposed a self-power supply device which is base on triboelectric nanogenerators and electromagnetic generator.” has been modified to “[32] proposes a self-power supply device which is base on triboelectric nanogenerators and electromagnetic generator.”
- On Page 2 Line 47:“[14] designed the triboelectric electromagnetic hybrid nanogenerator for self-powered sensors.”has been modified to“[14] designs the triboelectric electromagnetic hybrid nanogenerator for self-powered sensors.”
- On Page 2 Line 68:“When the external wind speed changes, the HSVA harvester can enhance the power generation efficiency of the generator to reduce the gap between actual and desired power by changing its own mechanical structure to match wind speed.”has been modified to“When the external wind speed changes, the HSVA harvester can enhance the power generation efficiency of the generator to decrease the power error by changing its own mechanical structure to match wind speed.”
- On Page 2 Line 75:“Additionally, the structure of the HSVA harvester has a strong protective effect as the wind speeds rose 10.1m/s.”has been modified to“Additionally, the structure of the HSVA harvester has a strong protective effect as the wind speed rises to 10m/s.”
- On Page 5 Line 170:“In addition, it also increases the replenishing breeze energy of the concave surface airflow on the side air side of the convex surface, and the positive torque has also increased.”has been modified to“In addition, it increases the replenishing breeze energy of the concave surface airflow on the side air side of the convex surface, and the positive torque also increases.”
- On Page 6 Line 190:“In a moderate breeze environment, the star-up wind speed of the HSVA harvester is 0.8m/s, are smaller than those when the HFA harvester is driven (1.2m/s).”has been modified to“In a moderate breeze environment, the star-up wind speed of the HSVA harvester is 0.8m/s, is smaller than the star-up wind speed of the HFA harvester (1.2m/s).”
- On Page 6 Line 211:“Therefore, as wind speed increased or decreased, the wind cup captures the random breeze energy of the environment directly, and the HSVA harvester can automatically adjust to typical of different winds.”has been modified to“Therefore, as wind speed increases or decreases, the wind cup captures the random breeze energy of the environment directly, and the HSVA harvester can automatically adjust to typical of different winds.”
- On Page 8 Line 231:“Moreover, as in environment of strong breeze, the function of helix s-type vertical axis can protect the structure of the HSVA harvester avoiding damage.”has been modified to“Moreover, in environment of strong breeze, the function of helix s-type vertical axis can protect the structure of the HSVA harvester avoiding damage.”
- On Page 8 Line 239:“On Fig. 7b, when wind speed is 4.1 m/s, the voltage trends are charged with different capacitors.”has been modified to“In Fig. 7b, when the wind speed is 4.1 m/s, the voltage are charged with different capacitors.”
- On Page 9 Line 253:“Additionally, in Fig. 7d and 7e, the HSVA harvester is applied to power a wireless hygrothermograph, which can sense the surrounding temperature as well as humidity, and the data can be send to the smart phone though the bluetooth technology.”has been modified to“Additionally, in Fig. 7d and 7e, the HSVA harvester is applied to power a wireless hygrothermograph, which can sense the surrounding temperature as well as humidity, and the data is transmitted to the smart phone though the bluetooth technology.”
- On Page 9 Line 275:“the HSVA harvester is well suited for the exploitation of natural random wind that normally has weak driving force, varied speed, and intermittent availability.”has been modified to“The HSVA harvester is well suited for the exploitation of natural random wind that normally has weak driving force, varied speed, and intermittent availability.”

Reviewer 2 Report
The authors have proposed a harvester with helix s-type vertical axis to capture random breeze energy efficiently. This paper has concerns in terms of novelty and validation.
1. The literature review is limited. Thus, it is recommended to explore the latest literature.
2. Problem objectives are not clear, Author should mention all objectives with the gaps.
3. A detailed comparative analysis is required.
4. This reviewer is unable to find the novelty. Moreover, the scheme was not supported by the hardware/experimental/real-time/HIL/prototype results.
The language of the text is generally grammatically correct.
Author Response
Response to reviewer
Reviewer: 2
The authors have proposed a harvester with helix s-type vertical axis to capture random breeze energy efficiently. This paper has concerns in terms of novelty and validation.
- The literature review is limited. Thus, it is recommended to explore the latest literature.
Response:
Several latest literature about energy harvesting published in this journal have been cited.
- Problem objectives are not clear, Author should mention all objectives with the gaps.
Response:
All objectives with the gaps have been mentioned in the revised manuscript. There are three problem objectives: Firstly, the HSVA harvester can match the random wind to achieve highly efficient power in the natural breeze environment. Secondly, the HSVA is constructed by two semi-circular buckets driven by the difference of the drag exerted, and the proposed harvester achieves a low start-up wind speed of 0.8 m/s. Finally, after the wind stops, the HSVA harvester can work with longer duration than the HFA harvester under the same time. The experimental results verify the effectiveness of the proposes HSVA harvester.
- A detailed comparative analysis is required.
Response:
Several detailed comparative analyses are at Section 3 in the revised manuscript.
- This reviewer is unable to find the novelty. Moreover, the scheme was not supported by the hardware/experimental/real-time/HIL/prototype results.
Response:
In order to better validate the novelty and applicability of the proposed HSVA, an experimental application platform has been built. Multiple experimental result figures have been generated and two videos have been recorded. The experimental results verify the effectiveness of the proposes HSVA harvester. The output peak power can reach 17.1 mW at the wind speed of 10.1 m/s that is 2.95 times higher as compared with the corresponding harvester with the HFA harvester. Furthermore, as the wind speed gradually increases, the proposed HSVA harvester achieves a low start-up wind speed of 0.8 m/s. Moreover, experiments showed that the operating time of the HSVA harvester was 44.6s longer than that of the HFA harvester after the wind stopped from 4.5 m/s. The HSVA harvester is well suited for the exploitation of natural random wind that normally has weak driving force, varied speed, and intermittent availability. Therefore, the HSVA harvester can serve as a green energy source for some sensors to monitor the environment.

Reviewer 3 Report
The breeze energy harvesting/capturing system construction is an interesting yet unpopular topic for researchers. Here the authors report their own design with the specific working performance, which of course deserves publication in this journal. Some issues are supposed to be taken care of before the formal acceptance:
1. The discussion part can include that wind energy is generated by temperature difference, which is, in essence, a derived form of solar power. Thus, some solar energy researches can be cited as a support: Energy Environ. Sci. 2023, 16 (5), 2316-2326.; Adv. Mater. 2023, 2304632.; Adv. Mater. 2023, 35 (18), 2212275.
2. The energy-converting efficiency from wind to electricity is meaningful and eye-catching. Please make a Table to demonstrate the specific efficiency values for each application scenario.
3. The wind energy calculation part: Is there any boundary condition considered? and do author need to add some correlated equation?
4. In my own opinion, Fig 3 can be merged into Fig 1.
It's generally OK, but an overall spell checking is always suggested.
Author Response
Response to reviewer
Reviewer: 3
The breeze energy harvesting/capturing system construction is an interesting yet unpopular topic for researchers. Here the authors report their own design with the specific working performance, which of course deserves publication in this journal. Some issues are supposed to be taken care of before the formal acceptance:
- The discussion part can include that wind energy is generated by temperature difference, which is, in essence, a derived form of solar power. Thus, some solar energy researches can be cited as a support: Energy Environ. Sci. 2023, 16 (5), 2316-2326.; Adv. Mater. 2023, 2304632.; Adv. Mater. 2023, 35 (18), 2212275.
Response:
Thank you for your suggestion, some solar energy researches published have been cited.
- The energy-converting efficiency from wind to electricity is meaningful and eye-catching. Please make a Table to demonstrate the specific efficiency values for each application scenario.
Response:
We have added Table 1 to illustrate the relationship between wind speed and out power of two different harvesters. It is further proved that the HSVA harvester has better power conversion efficiency under different random wind speeds after the two harvesters start. As the wind speed increases gradually, the gap in conversion efficiency is widening between two harvesters.
- The wind energy calculation part: Is there any boundary condition considered? and do author need to add some correlated equation?
Response:
In this manuscript, the wind speed can be divided into three types: moderate wind (0m/s - 4,5m/s), fresh wind (4.5m/s - 8m/s) and strong wind (8m/s - 12m/s). Based on the mechanical structure and characteristics of the harvesters, the boundary conditions regarding wind speed have been taken into account. The boundary conditions of the HFA range from the start-up wind speed (1.2m/s) to the maximum rotation speed (396 rpm). The boundary conditions of the HSVA range from the start-up wind speed (0.8m/s) to the maximum rotation speed (685 rpm). If the rotation speed exceeds the maximum speed, the conversion power could no longer increase, and even the structure of the harvesters may be damaged.
- In my own opinion, Fig 3 can be merged into Fig 1.
Response:
Thanks a lot for your suggestion, the Fig 3 has been merged into Fig 1 in the revised manuscript.

Round 2
Reviewer 1 Report
No response letter is found, and please answer the questions one by one.
Minor editing of English language required.
Reviewer 2 Report
The authors have addressed all my queries. Thus, I recommend accepting the article for publication.
The paper can be accepted.
Round 3
Reviewer 1 Report
They have answered my questions.
Professional English polishing is needed.